# Zinc Iodide-Metal Chloride-Organic Base: An Efficient Catalytic System for Synthesis of Cyclic Carbonates from Carbon Dioxide and Epoxides under Ambient Conditions

**Masatoshi Mihara \*, Shuichi Nakao, Takeo Nakai and Takumi Mizuno**

Osaka Research Institute of Industrial Science and Technology, 1-6-50 Morinomiya, Joto-ku, Osaka 536-8553, Japan
* Correspondence: mihara@omtri.or.jp

**Abstract:** Development of an effective catalytic system for the cycloaddition of carbon dioxide to epoxides for the preparation of cyclic carbonates under mild conditions is of great importance. Herein, a mixture of zinc iodide, metal chlorides, and strong organic bases is demonstrated to be a useful catalytic system that works at room temperature under atmospheric pressure. The most efficient combination, zinc iodide-niobium chloride-7-methyl-1,5,7-triazabicyclo[4.4.0]dec-5-ene (1.2-0.3-3.0 mol%), gave styrene carbonate (95%) from styrene oxide and $CO_2$ (balloon) at 25 °C for 24 h. Another combination, zinc iodide-zinc chloride-1,8-diazabicyclo[5.4.0]undec-7-ene (1.2-0.8-4.0 mol%), kept the catalytic activity for the preparation of propylene carbonate until the fourth run. Therefore, the reaction system was operationally simple, highly efficient, and proceeded under ambient conditions. The catalyst is composed of readily available reagents and is reusable. Thus, the method presented is a powerful tool for utilizing $CO_2$ as the starting material for the production of valuable chemicals.

**Keywords:** carbon dioxide; epoxides; cyclic carbonates

## 1. Introduction

$CO_2$ is regarded as an ideal carbon source for organic synthesis because it is inexpensive, abundant, nontoxic, and renewable. Therefore, efficient transformation of $CO_2$ into valuable chemicals is important for creating greener and more sustainable industries. The cycloaddition of $CO_2$ and epoxides to afford five-membered cyclic carbonates is one of the most promising methods for utilizing $CO_2$, owing to the 100% atom economy of the reaction. Carbonates are widely used as polar aprotic solvents, electrolytes in lithium-ion batteries, and intermediates in the production of pharmaceuticals and fine chemicals. Accordingly, a wide range of methods involving metal-based catalysts and organocatalysts have been reported for the synthesis of cyclic carbonates using $CO_2$ [1–3]. For example, the catalytic components employed in this study—zinc halides [4–19], NbCl$_5$ [20–24], and 7-methyl-1,5,7-triazabicyclo[4.4.0]dec-5-ene (MTBD) [25]—have been previously used as catalysts for the cycloaddition of $CO_2$, often in combination with onium salts. In addition, zinc-based heterogeneous catalysts have been continuously reported [4–6,8,9,11–14,17–19,26–30]. Despite the extensive research in this area, pressurizing or/and heating are usually required to obtain cyclic carbonates from $CO_2$ and epoxides in satisfactory yields.

Owing to the growing environmental and energy concerns, the development of synthetic methodologies for the preparation of cyclic carbonates by harnessing $CO_2$ under mild conditions is gaining interest. Recently, metal complexes [31–42], metal-organic frameworks [43–45], and solid-supported reagents [46–52] have been reported for facile conversion of $CO_2$ and epoxides into cyclic carbonates, even under ambient conditions (around 25 °C under 0.1 MPa). However, these materials typically have complex chemical structures with relatively high molecular weights. Organocatalysts have been utilized under ambient conditions; however, they usually require loadings greater than 5 mol% [53–59].

These limitations have spurred catalytic research toward the development of more effective and practical methods for the synthesis of cyclic carbonates under milder conditions.

We recently reported the selective guanidine-accelerated synthesis of carbonates from $CO_2$, glycerol, and alkyl halides [60]. In this study, we observed that a zinc complex formed between $ZnI_2$ and guanidine displayed catalytic activity for the cycloaddition of $CO_2$ and epoxides. Continuing our investigations in this area of research, we report the development of an efficient catalytic system for the conversion of $CO_2$ to cyclic carbonates using $CO_2$ at room temperature and atmospheric pressure. The catalyst was developed using simple metal halides and strong organic bases, and was catalytically active under ambient conditions.

## 2. Results

Synthesis of styrene carbonate, **2a**, from styrene oxide, **1a**, and $CO_2$ was investigated as a model reaction in the presence of metal halides and guanidine (Table 1). The combination of $ZnI_2$ and pentaalkylguanidine led to a 26% conversion to **2a** (entry 1). In contrast, the addition of $ZnCl_2$ as a cocatalyst substantially improved the conversion (entry 2). Catalytic activity almost completely disappeared in the absence of $ZnI_2$ and/or guanidine (entries 3 and 4). Thus, the combination of $ZnI_2$, $ZnCl_2$, and guanidine enhances the catalytic activity, indicating the presence of synergistic effects that contribute to the acceleration of the reaction.

**Table 1.** Cycloaddition of styrene oxide (**1a**) and $CO_2$ in the presence of zinc halides and guanidine [a].

| Entry | Catalyst | Cocatalyst | Conversion (%) [b] |
|---|---|---|---|
| 1 | $ZnI_2/t\text{BuN=C(NMe}_2)_2$ | none | 26 |
| 2 | $ZnI_2/t\text{BuN=C(NMe}_2)_2$ | $ZnCl_2$ | 81 |
| 3 | $t\text{BuN=C(NMe}_2)_2$ | $ZnCl_2$ | 1 |
| 4 | none | $ZnCl_2$ | 0 |

[a] Reaction conditions: **1a** (10 mmol), $CO_2$ (0.1 MPa), $ZnI_2/t\text{BuN=C(NMe}_2)_2$ (1.0/2.2 mol%), $ZnCl_2$ (0.5 mol%), 25 °C, 18 h; $ZnI_2/t\text{BuN=C(NMe}_2)_2$ was prepared by mixing $ZnI_2$ and $t\text{BuN=C(NMe}_2)_2$ in ethanol followed by solvent removal. [b] Determined by $^1$H-NMR area ratio.

The catalyst combination was further optimized by evaluating several metal chlorides, organic bases, and metal iodides to replace $ZnCl_2$, *N,N′,N′,N″,N″-tert*-butyltetramethylguanidine, and $ZnI_2$, respectively (Table 2). The effect of the metal chlorides was first examined under the conditions described in entry 2 of Table 1 (combination A). Although all metal chlorides enhanced the catalytic reaction, their effects were significantly different. $NbCl_5$ afforded the best conversion (83%), which might be due to the suitable Lewis acidity of $NbCl_5$ for the cycloaddition in the presence of guanidines and $ZnI_2$. The effect of organic bases was then examined using $NbCl_5$ as a metal halide (combination B). The results indicated that a strong basicity was necessary to achieve good conversion, which was further evidenced by the low conversion observed for imidazole. Furthermore, the presence of sterically demanding substituents on the guanidines did not affect the cycloaddition conversion, and MTBD demonstrated the best conversion (86%). The effect of the metal iodide was further examined using $NbCl_5$ and MTBD (combination C). $ZnI_2$ was found to be the best iodide anion source despite its low number of iodine atoms per metal center. The catalytic activity of $ZnI_2$ was higher than that of $ZnBr_2$, which can be attributed to the facile generation of the iodide anion from $ZnI_2$ compared with the generation of the bromide anion from $ZnBr_2$. Additionally, although $NbCl_5$-$Bu_4NBr$ (0.5–1 mol%) was reported to be an effective catalyst under mild conditions (45 °C) [20],

this catalytic system did not deliver good results (conversion: 11%) under the present conditions (NbCl$_5$-Bu$_4$NBr: 0.5–2 mol%, 25 °C). Our optimization results indicate that ZnI$_2$-NbCl$_5$-MTBD is the best combination, which probably leads to effective cooperative catalysis of acidic sites and iodide anions.

**Table 2.** Metal iodide/base and metal chlorides-catalyzed synthesis of styrene carbonate (**2a**) from styrene oxide (**1a**) and CO$_2$ [a].

1a + CO$_2$ $\xrightarrow{\text{metal iodide/base + metal chloride}}$ 2a

| Combination A: ZnI$_2$/$t$BuN=C(NMe$_2$)$_2$ + metal chloride | | | | | | |
|---|---|---|---|---|---|---|
| Metal chloride | NbCl$_5$ | ZnCl$_2$ | ZrCl$_4$ | BiCl$_3$ | FeCl$_3$ | MgCl$_2$ | CuCl$_2$ |
| Conversion [b] (%) | 83 | 81 | 75 | 75 | 73 | 71 | 41 |

| Combination B: ZnI$_2$/base + NbCl$_5$ | | | | | | |
|---|---|---|---|---|---|---|
| Base [c] | MTBD | DBU | $t$BuG | $n$BuG | TBD | PhG | 1,2-dimethylimidazole |
| Conversion [b] (%) | 86 | 84 | 83 | 83 | 80 | 65 | 22 |

| Combination C: Metal iodide (bromide)/MTBD + NbCl$_5$ | | | | | |
|---|---|---|---|---|---|
| Metal iodide (bromide) | ZnI$_2$ | SnI$_4$ | TiI$_4$ | BiI$_3$ | ZnBr$_2$ |
| Conversion [b] (%) | 86 | 80 | 55 | 50 | 44 |

[a] Reaction conditions: **1a** (10 mmol), CO$_2$ (0.1 MPa), metal iodide/base (1.0/2.2 mol%), metal chloride (0.5 mol%), 25 °C, 18 h. [b] Determined by $^1$H-NMR area ratio. [c] DBU: 1,8-diazabicyclo[5.4.0]undec-7-ene, G: N=C(NMe$_2$)$_2$, TBD: 1,5,7-triazabicyclo[4.4.0]dec-5-ene.

ZnI$_2$ and MTBD complexes were prepared prior to NbCl$_5$ addition, and the catalytic system prepared by this process was used for the cycloaddition reaction. However, the simultaneous mixing of ZnI$_2$, MTBD, and NbCl$_5$ did not affect product conversion (Table 3). Therefore, NbCl$_5$ was found to be effective upon the direct addition of free guanidine. We further observed that the conversion depended significantly on the ZnI$_2$, MTBD, and NbCl$_5$ mole ratio. The use of a suitable ratio led to almost complete consumption of the starting material with a prolonged reaction time [ZnI$_2$:MTBD:NbCl$_5$ = 1.2:3.0:0.3, reaction time (24 h), conversion (99%), and isolated yield (95%)]. Overall, the optimized method is operationally simple, does not require the preformation of the catalyst, and allows the efficient cycloaddition of CO$_2$ and epoxides under ambient conditions.

**Table 3.** Comparison of conversions to styrene carbonate (**2a**) with and without the preformed ZnI$_2$/MTBD [a].

1a + CO$_2$ $\xrightarrow{\text{ZnI}_2,\ \text{MTBD},\ \text{NbCl}_5}$ 2a

| Catalyst [b] | Conversion [c] (%) |
|---|---|
| ZnI$_2$/MTBD + NbCl$_5$ [d] | 86 |
| ZnI$_2$-MTBD-NbCl$_5$ [e] | 86 |

[a] Reaction conditions: 1a (10 mmol), CO$_2$ (0.1 MPa), 25 °C, 18 h. [b] ZnI$_2$ (1.0 mol%), MTBD (2.2 mol%), NbCl$_5$ (0.5 mol%). [c] Determined by $^1$H-NMR area ratio. [d] After the preformation of ZnI$_2$/MTBD, NbCl$_5$ was added. [e] ZnI$_2$, MTBD, and NbCl$_5$ were added simultaneously.

The optimized conditions were used for the conversion of several epoxides in the presence of CO$_2$, and the corresponding five-membered cyclic carbonates were isolated in moderate to excellent yields (Table 4). Various terminal epoxides bearing aryl, alkyl, ether, chloro, long alkyl chains, and unsaturated groups reacted smoothly with CO$_2$ (atmospheric pressure) at room temperature to furnish the corresponding carbonates in high yields (entries 1–6). The bis-epoxide was also converted to the corresponding bis-cyclic carbonate in excellent yield (entry 7). However, the use of *N*-methylpyrrolidone (NMP) as a solvent was required to homogenize the reaction mixture in cases where the product precipitation

lowered the yield by hindering continuous stirring of the reaction mixture. Accordingly, the $ZnI_2$-$NbCl_5$-MTBD-catalyst system allowed the selective cycloaddition of $CO_2$ with terminal epoxides under solvent-free conditions and proceeded without by-product formation. However, internal epoxides did not afford the desired carbonates under ambient conditions. Nevertheless, the desired cycloaddition was achieved by increasing the reaction temperature and $CO_2$ pressure (entries 8 and 9). Remarkably, even a sterically hindered internal epoxide was converted to the corresponding carbonate product in moderate yield.

**Table 4.** Efficient synthesis of a cyclic carbonate (**2**) under mild conditions [a].

| Entry | $R^1$ | $R^2$ | 2 | Yield (%) [b] |
|-------|-------|-------|---|---------------|
| 1 | Ph | H | **2a** | 95 |
| 2 | Me | H | **2b** | 90 |
| 3 [c] | $CH_2OPh$ | H | **2c** | 92 |
| 4 | $CH_2Cl$ | H | **2d** | 85 |
| 5 [c] | $C_8H_{15}$ | H | **2e** | 89 |
| 6 | $CH_2OCH_2CH=CH_2$ | H | **2f** | 89 |
| 7 [d] | $(CH_2)_4$ | H | **2g** | 95 |
| 8 [e] | $(CH_2)_4$ | | **2h** | 38 [f] |
| 9 [e] | Ph (*trans*) | Ph | **2i** | 53 [g] |

[a] Reaction conditions: epoxides (10 mmol), $CO_2$ (0.1 MPa), $ZnI_2$ (1.2 mol%), $NbCl_5$ (0.3 mol%), MTBD (3.0 mol%), 25 °C, 24 h. [b] Isolated yields. [c] NMP (1 mL) was added. [d] $ZnI_2$ (2.4 mol%), $NbCl_5$ (0.6 mol%), MTBD (6.0 mol%). [e] $CO_2$ (1.0 MPa), $ZnI_2$ (4.0 mol%), $NbCl_5$ (1.0 mol%), MTBD (10.0 mol%), 120 °C. [f] Mixture of *cis/trans* (84/16). [g] Only *trans*.

After studying the scope of the catalytic reaction, we evaluated the reusability of the catalyst system. The reusability of $ZnI_2$-$NbCl_5$-MTBD was assessed using the reaction of propylene oxide (**1b**) with $CO_2$ (Table 5). After the first experiment, the product was isolated by the distillation of the crude mixture because of the homogeneous system, and the residual catalyst in the reaction vessel was used for two subsequent runs under the same reaction conditions without any pretreatment. Unfortunately, a slight decrease in yield was observed in the second run, which was maintained in the third run (catalyst A). The reduction in catalytic efficiency can be attributed to the partial decomposition of $NbCl_5$ and MTBD during distillation. Therefore, we further examined the cycloaddition using a metal chloride and an organic base, which are less moisture- and heat-sensitive under similar conditions. The use of $ZnCl_2$ and DBU instead of $NbCl_5$ and MTBD led to similar conversions during the recycling experiments, although larger quantities of the reagents were required (catalyst B). Thus, the $ZnI_2$-$ZnCl_2$-DBU system is a highly stable cycloaddition catalyst that does not display any significant decrease in catalytic activity until the fourth run.

**Table 5.** Reuse of catalyst in the synthesis of propylene carbonate (**2b**) from propylene oxide (**1b**) and $CO_2$ [a].

$$\textbf{1b} \; + \; CO_2 \xrightarrow{\text{catalyst A or B}} \textbf{2b}$$

| Catalyst A: $ZnI_2$-$NbCl_5$-MTBD [b] | | |
|:---:|:---:|:---:|
| Cycle | 1 | 2 | 3 |
| Yield (%) | 84 | 67 | 68 |

| Catalyst B: $ZnI_2$-$ZnCl_2$-DBU [c] | | | |
|:---:|:---:|:---:|:---:|
| Cycle | 1 | 2 | 3 | 4 |
| Yield (%) | 78 | 81 | 85 | 83 |

[a] Reaction conditions: **1b** (10 mmol), $CO_2$ (0.1 MPa), 25 °C, 24 h. [b] $ZnI_2$ (1.2 mol%), $NbCl_5$ (0.3 mol%), MTBD (3.0 mol%). [c] $ZnI_2$ (1.2 mol%), $ZnCl_2$ (0.8 mol%), DBU (4.0 mol%).

## 3. Discussion

The catalytic system was examined by $^1$H-NMR in DMSO-$d_6$ using tetramethylsilane as an internal standard. The addition of $ZnI_2$ or $NbCl_5$ to the MTBD led to a downfield shift in the peaks corresponding to the MTBD (Figure 1A). In addition, the addition of $NbCl_5$ to the MTBD and $ZnI_2$ resulted in a further downfield shift in the MTBD peaks. These changes are indicative of MTBD coordination to $ZnI_2$ and $NbCl_5$. Moreover, the addition of styrene oxide (**1a**) to the complexes led to a slight downfield shift of the **1a** signals, which suggests coordination of the formed complexes with **1a** (Figure 1B). However, because bubbling $CO_2$ into the mixture did not cause changes in the chemical shifts, the MTBD complex coordinated with $ZnI_2$ and $NbCl_5$ might be less able to interact with $CO_2$. Furthermore, the coordination of $ZnI_2$ and $NbCl_5$ to MTBD was also confirmed by the changes in the IR spectra (Figure S1). Accordingly, considering the key contribution of iodide to the catalyst activity (almost no reaction occurred in the absence of $ZnI_2$, Table 1, entries 2 vs. 3), the main catalytic effect should operate via the synergistic activation of epoxides by the acidity of the niobium and zinc complexes, followed by the nucleophilic attack of the iodide anion. The proposed catalytic cycle for the synthesis of cyclic carbonates is shown in Scheme 1 [61–63]. The epoxide ring was activated by coordination with the Nb and Zn acidic sites. The iodide anion generated from the reaction between $ZnI_2$ and guanidines simultaneously attacks the epoxide ring to form an intermediate [I] (path A). The other path B, including the attack of MTBD followed by exchanging the MTBD moiety by the iodide, can also be proposed. Nucleophilic addition of the alkoxide to $CO_2$ forms a hemicarbonate intermediate [II]. Because the iodide anion is a good leaving group, the ring closure of the metal carbonate occurs smoothly, affording the desired cyclic carbonate.

Previous reports on styrene carbonate synthesis revealed that the preceding combination of metal halides and organic bases afforded good yields under severe conditions. For example, $SnCl_4$ [64] or $ZnI_2$ [5,13,15] were employed in combination with bases under 0.3 MPa-75 °C or 1–3 MPa-60–150 °C, respectively. Additionally, $ZnI_2$ [6–8] or $NbCl_5$ [24] in combination with base-HX also were used under heating and pressurized $CO_2$ conditions (30–130 °C, 1–2 MPa). Therefore, the higher efficiency reported herein supports the occurrence of cooperative catalysis by $ZnI_2$, $NbCl_5$, and MTBD. In this catalytic system, the coordination of $ZnI_2$ and $NbCl_5$ to MTBD may hold $I^-$ derived from $ZnI_2$ closer to the Lewis acids and increase the solubility of their metal halides in the reaction mixture, thus enhancing the activity of this reaction. Furthermore, although several effective catalysts for the preparation of styrene carbonate, even under ambient conditions, have already been reported [31–42,44–46,48–55,58,59], they usually require a long time for multistep preparation (overnight to 30 days, 1–3 steps), simultaneous use of a relatively high-loading tetrabutylammonium halide (1–50 mol%), and/or the use of significant quantities despite low catalytic loadings (73–1737 mg for 10 mmol of styrene oxide). However, our reaction allows convenient coupling with a relatively low-loading catalyst composed of readily available reagents ($ZnI_2$-$NbCl_5$-MTBD: 92 mg for 10 mmol). In addition, while TON (mol of product/mol of metal) and TOF [TON/time (h)] of previous catalysts cited here are 16–2760

and 0.7–58 for the production of styrene carbonate, respectively, those of our one are 63 and 2.6, respectively. Thus, our catalytic system shows near average values among them except the highest TON and TOF [46] attained in the presence of tetrabutylammonium bromide (7.2 mol%) as a cocatalyst.

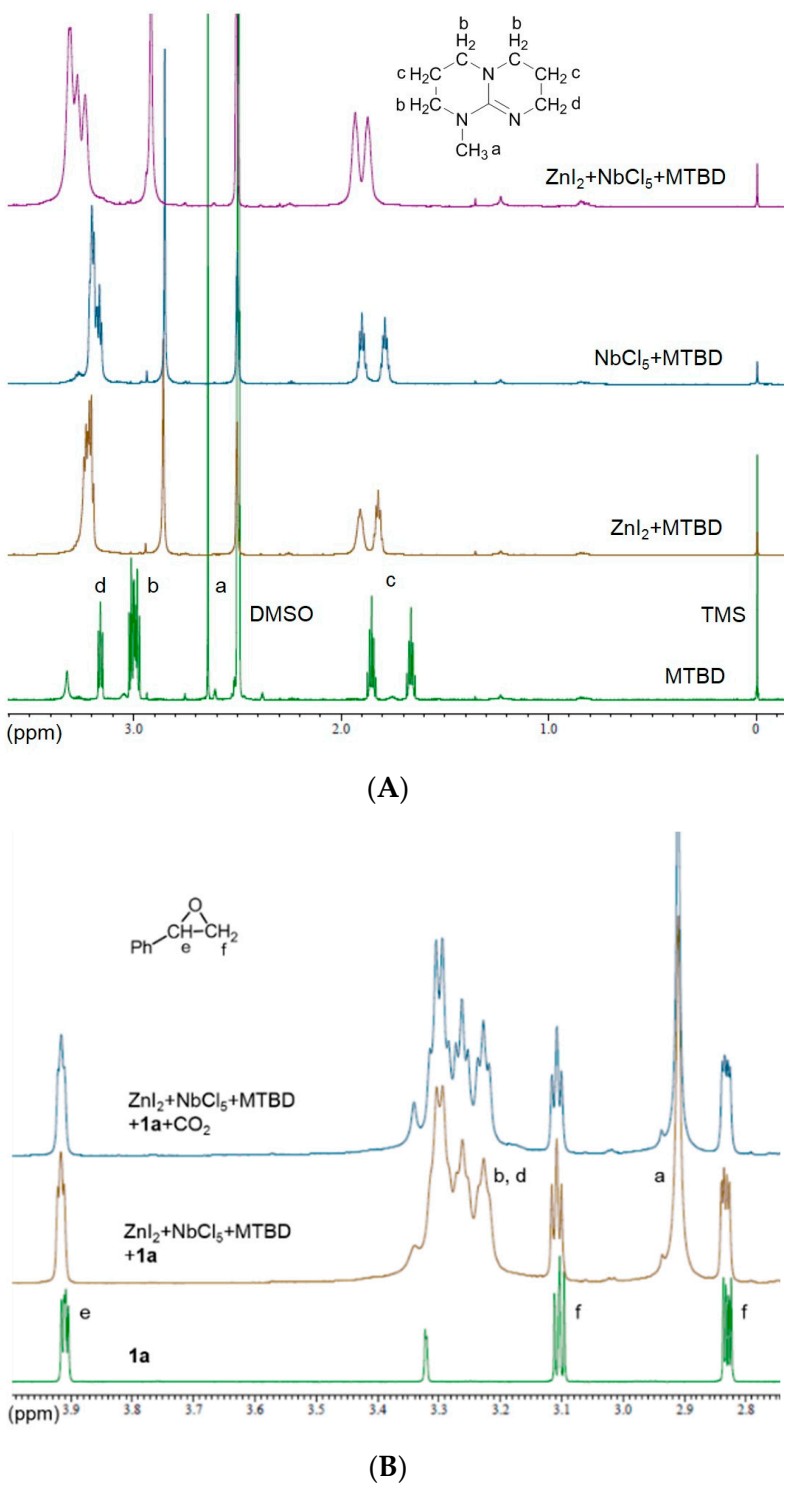

**Figure 1.** $^1$H-NMR of a mixture of catalysts, styrene oxide (**1a**), and $CO_2$; (**A**) $ZnI_2$ (0.04 mmol), $NbCl_5$ (0.01 mmol), and MTBD (0.1 mmol); (**B**) $ZnI_2$-$NbCl_5$-MTBD, **1a** (0.1 mmol), and $CO_2$.

LA-B I⁻ = (ZnI₂)ₚ-(NbCl₅)_q-(MTBD)ᵣ

**Scheme 1.** Plausible mechanism for the cycloaddition of styrene oxide and $CO_2$.

## 4. Materials and Methods

### 4.1. Materials

$ZnI_2$, $ZnCl_2$, $NbCl_5$, DBU, MTBD, epoxides, deuterated solvents, and $CO_2$ (99.5%) were used without further purification. NMP (super dehydrated) and EtOH (super dehydrated), purchased from Wako Pure Chemical Industries Ltd., were used without further treatment.

### 4.2. Methods

A general procedure for $ZnI_2$/organic base + metal chloride-catalyzed synthesis of styrene carbonate (**2a**) from styrene oxide (**1a**) and $CO_2$

$ZnI_2$ (0.1 mmol), organic bases (0.22 mmol), EtOH (0.4 mL), and a magnetic stirring bar were placed in a glass vessel connected to an injection port with a 3-way cock. The mixture was stirred at 60 °C for 3 h and was evaporated around 25 °C under reduced pressure to afford the $ZnI_2$/organic base complex. To the complex, metal chlorides (0.05 mmol) and **1a** (10 mmol) were added, and the vessel was charged with $CO_2$ from a balloon. The reaction mixture was stirred at 25 °C for 24 h under a 0.1 MPa pressure of $CO_2$. The conversions to the desired carbonates were determined based on the [1]H NMR area ratios (**2a**/**1a**+**2a**).

A general procedure for $ZnI_2$-$NbCl_5$-MTBD-catalyzed synthesis of cyclic carbonates **2** from epoxides **1** and $CO_2$

$ZnI_2$ (0.12 mmol), MTBD (0.3 mmol), $NbCl_5$ (0.03 mmol), and **1** (10 mmol) were placed with a magnetic stirring bar in a glass vessel connected to an injection port with a 3-way cock. NMP was added as the solvent in the experiments for glycidyl phenyl ether (**1c**) and 1,2-epoxydecane (**1e**). The vessel was charged with $CO_2$ from a balloon, and the reaction mixture was stirred at 25 °C for 24 h under a 0.1 MPa pressure of $CO_2$. The obtained crude product was purified by silica gel column chromatography (silica gel 60) to isolate **2**. The products **2b** and 4-chloromethyl-1,3-dioxolan-2-one (**2d**) were purified by distillation instead of purification on silica gel column chromatography.

A procedure for reuse of $ZnI_2$-$NbCl_5$-MTBD and $ZnI_2$-$ZnCl_2$-DBU in the synthesis of propylene carbonate (**2b**) from propylene oxide (**1b**) and $CO_2$

The carbonate **2b** was synthesized from **1b** and $CO_2$ according to the procedure described above, using $ZnI_2$-$NbCl_5$-MTBD or $ZnI_2$-$ZnCl_2$-DBU. The crude **2b** was purified by distillation under reduced pressure to obtain pure **2b**. To the residual catalyst in the reaction vessel, fresh **1b** (10 mmol) was added, $CO_2$ was charged from a balloon, and the reaction mixture was stirred at 25 °C for 24 h in next recycle experiment. The resulting liquid was purified by distillation to afford **2b**, and the residual catalyst was further used for the third run.

A procedure for [1]H NMR measurement to examine the reaction mechanism.

$ZnI_2$ (0.04 mmol), MTBD (0.1 mmol), $NbCl_5$ (0.01 mmol), and DMSO-d6 + 0.03% TMS (0.75 mL) were placed in an NMR tube, for studying the coordination of metal halides to

MTBD. The $^1$H NMR spectrum was acquired. After that, **1a** (0.1 mmol) was added to the complex, and $^1$H NMR was acquired again. $CO_2$ was bubbled into the mixture, and the $^1$H NMR was acquired again.

**5. Conclusions**

The $ZnI_2$-$NbCl_5$-MTBD-catalyzed synthesis of carbonates is a powerful tool for utilizing $CO_2$ under extremely mild conditions. The catalytic system allowed the conversion of various terminal epoxides into the corresponding cyclic carbonates in high yields (85–95%) with a relatively low-loading catalyst (92 mg for 10 mmol of epoxides) under ambient conditions. The catalyst was also prepared in situ by simply mixing readily available reagents and could be used under solvent-free conditions; thus, it is a practical and environmentally benign reagent. To the best of our knowledge, no catalysts composed of two different metal halides or organic bases have been reported for the synthesis of carbonates. As has already been observed for some metal-organic framework catalysts [65,66], the use of a dual Lewis acid catalytic system in combination with nucleophiles is believed to be a promising strategy for the development of synergistic catalysts for the cycloaddition of $CO_2$.

**Supplementary Materials:** The following supporting information can be downloaded at: https://www.mdpi.com/article/10.3390/catal13081214/s1, Characterization of Compounds **2a–i** [34,67–69], Figure S1: IR spectra of MTBD, $ZnI_2$+MTBD, $NbCl_5$+MTBD, and $ZnI_2$+$NbCl_5$+MTBD.

**Author Contributions:** Conceptualization, M.M. and T.M.; methodology, T.N., S.N. and M.M.; formal analysis, T.N., S.N. and M.M.; investigation, M.M.; data curation, M.M.; writing—original draft preparation, M.M.; writing—review and editing, M.M.; supervision, T.M.; project administration, M.M.; funding acquisition, M.M., T.N. and T.M. All authors have read and agreed to the published version of the manuscript.

**Funding:** This research was funded by JSPS KAKENHI Grant Number 17K05959.

**Data Availability Statement:** Not applicable.

**Conflicts of Interest:** The authors declare no conflict of interest.

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
