# Peer review of "Zinc Iodide-Metal Chloride-Organic Base: An Efficient Catalytic System for Synthesis of Cyclic Carbonates from Carbon Dioxide and Epoxides under Ambient Conditions"

_catalysts, doi:10.3390/catal13081214_

Round 1

Reviewer 1 Report

I suggest to revise the paper in https://doi.org/10.1016/j.mcat.2019.110637, to complete the background and compare methodologies.

Author Response

We have studied their comments carefully and have made the corrections requested.  I am returning herewith the revised manuscript with changes highlighted in red.  I thank you for your kind consideration in advance.

Reviewer 2 Report

In this manuscript, Mihara et al. report an efficient catalytic system based on triple-component complex (ZnI2-NbCl5-MTBD) for the cycloaddition of carbon dioxide to epoxides under mild conditions. The effect of different metals, bases and different ratios of components were investigated. In my opinion, the obtained here results are interesting for the readership of Catalysts. The manuscript can be accepted after addressing the next comments:

1. Unfortunately, I couldn’t find the SI file where the characterization of the obtained compounds should be described.

2. It is recommended to provide the results with different metal salts, additives, etc. in the separate tables to show the data more clear, and not only in the text.

3. The footnote in Scheme 1 better to give with letter and not with number (otherwise it can be considered as a refence on literature). The same is for Scheme 2, Table 2.

4. Which yields are given in Table 2 (isolated? By NMR?)

5. What is the reason for the increased yields for 2-4 runs in the case of Catalyst B (Table 3)?

6. The next papers dealt with zinc based heterogeneous catalysts can be cited (Chem. Eur. J. 2020, 26, 13686-13697; Appl. Catal. B 2019, 254, 380-390; ChemCatChem 2019, 11, 511-519; J. CO2 Utilization 2021, 53, 101718, J. Catal. 2022, 407, 65-76).

7. The next papers dealt with the catalysts operating under ambient conditions can be cited (Asian J. Org. Chem. 2022, 11, e202100811; Catal. Today 2020, 375, 324–334; Angew. Chem. Int. Ed. 2020, 59, 5830–5836; Inorg. Chem. Front. 2021, 8, 3871–3884).

Moderate editing of English language required

Author Response

(The authors gave the same response as above.)

Reviewer 3 Report

The authors prepared zinc-metal chloride-organic base as catalyst for CO2 conversion. The manuscript requires major revision:

1- The catalysts were just characterized by 1H NMR. Further characterization (ICP, FT-IR, etc.) is required to prove the coordination of ZnI2 and NbCl5 to MTBD, etc.

2- The metal content of the recovered catalysts should be determined and compared with that of fresh catalyst.

3- It should be mentioned in the text that the catalytic system is homogeneous or heterogeneous.

4- Different steps for Scheme 3 is not described in the manuscript.

5- Please change Scheme 1 and Scheme 2. They just contain text and it is better to draw the structure of substrate and product.

6- Also, for Scheme 1, clarify metal chloride (for condition A), base (for condition B) and metal iodide (for condition C) in the image.

5- Using NbCl5 (for condition A) and using MTBD (for condition B) were reported to have the best conversion. However, the reason is missing. The authors should discuss it.

Author Response

(The authors gave the same response as above.)

Reviewer 4 Report

In this work, Mihara et al. report combinations of Lewis bases, ZnI2 and metal halides for the cycloaddition of CO2 to epoxides. The authors optimized the catalytic systems in order to attain high catalytic activity under ambient conditions. Whereas the work is not particularly novel because these kinds of catalysts were all reported in the past, my feeling is that this is a well carried out study that provides, after optimization, nicely active catalytic systems and can assist in the future design of advanced catalysts. At the same time, the work should require improvement, a much deeper consideration of previous literature and should be re-evaluated after major revisions as noted:

a) While some references in the introduction are correct, the discussion should be more comprehensive, accurate and the correct works should be given credit; for instance in the following sentences:

-"...For example, the catalytic components employed in this study — ZnI2 [4-10]...."; in reality, the discovery of ZnCl2 and ZnI2 as catalysts for the target cycloaddition reaction is much older than the references provided by the authors, perhaps even older than the authors themselves (Chem. Ber. 19861191090 1094) The correct citation should be provided.

-"..Despite extensive research in this area, harsh reaction conditions are still required to obtain cyclic carbonates from CO2 and epoxides in satisfactory yields..." This sentence is not correct, there surely are homogeneous and heterogeneous catalysts that catalyze the cycloaddition reaction quite efficiently under ambient conditions (Chem. - Eur. J. 2016226556 6563; Catalysis Today, 2021, 375, 324-334; for a more comprehensive overview kindly refer to: ACS Catalysis, 2018, 8, 419-450.) Ideally, it would be worth carrying out a comparison of turnover numbers and turnover frequencies between the catalytic system reported by the authors and state-of-the-art systems available in literature that are active under ambient conditions.

-(lines 36-39) "..Recently, metal complexes [17-24], metal-organic frameworks [25-27], and solid-supported reagents [28-34] have been reported for facile conversion of CO2 and epoxides into cyclic carbonates, even under ambient conditions. However, these materials typically have complex chemical structures with relatively high molecular weights." Whereas it is true that many catalytic systems for the target reaction have incredibly sophisticated structure and are unlikely to be applied in a real-case scenario, there are other simple, surface supported heterogeneous catalysts that are very easily prepared and are active under ambient conditions (Chemical Engineering Journal, 2021, 422, 129930, Appl. Catal. B, 2019, 254, 380-390; for an overview see also Green Chemical Engineering, 2022, 3, 210-227). Additionally, there are already other extremely simple catalytic systems based on metal oxides that are (partly) recoverable and are active under atmospheric pressure (Inorg. Chem. 2022, 61, 41, 16356–16369; Chemical Engineering Research and Design, 2023, 191, 630-645). These aspects should be considered more carefully.

b) In lines 156-169 the authors consider the recovery of the catalytic system. While that seems to work, it requires the distillation of the high boiling carbonate phase that is clearly a non sustainable or viable approach under a realistic scenario. Recent examples have shown that molecular catalysts for the cycloaddition of CO2 to epoxides can be reused and recovered under biphasic reaction conditions (Green Chem., 2023, 25, 4336-4349). It would be useful for the readers if these concepts would be made available in the manuscript. 

c) At Page 5 of the manuscript the authors consider a possible reaction mechanism; it should be said that the proposed reaction mechanism is basically the same as earlier indicated for other NbCl5-based catalytic systems for the same or similar reactions (Catalysis Science & Technology, 2014, 4, 1534-1538; Molecular Catalysis, 2017, 443, 280-285). Going further into detail, recent research has shown that in catalytic systems for the target cycloaddition reaction that include a Lewis acidic metal and two nucleophiles (i.e. the halide anion and the Lewis base) a cooperative mechanism may occur where the two nucleophiles play different roles within the catalytic cycle (The Journal of Organic Chemistry, 2022, 87, 2873-2886). These concepts should be discussed in detail.

d) Line 198: "Nucleophilic addition of an alcohol to CO2 forms intermediate [II]." It would be much more correct to say that the nucleophilic attack of the alkoxide (not the alcohol) to CO2 leads to an hemicarbonate intermediate (II).

The English of the manuscript is ok

Author Response

(The authors gave the same response as above.)

Round 2

Reviewer 3 Report

The authors have made necessary revisions and I recommend the revised version of the manuscript for publication.

Author Response

We have made the corrections requested from academic editor and have confirmed all comments in the manuscript.  I am returning herewith the revised manuscript with changes highlighted in red.  I thank you for your kind consideration in advance.

Reviewer 4 Report

In this revised version I found that the authors correctly addressed the reviewers concerns and that the paper may be published as is.

Author Response

(The authors gave the same response as above.)
